# Covid-19's Impact on European Power Sectors: An Econometric Analysis

**Philipp Hauser** * [ID], **David Schönheit** [ID], **Hendrik Scharf** [ID], **Carl-Philipp Anke** [ID] **and Dominik Möst** [ID]

Chair of Energy Economics, Technische Universität Dresden, Münchner Platz 3, 01069 Dresden, Germany;
david.schoenheit@tu-dresden.de (D.S.); hendrik.scharf@tu-dresden.de (H.S.);
carl-philipp.anke@tu-dresden.de (C.-P.A.); dominik.moest@tu-dresden.de (D.M.)
* Correspondence: philipp.hauser@tu-dresden.de; Tel.: +49-0351-463-39680

**Abstract:** Covid-19 affects the personal lives of millions and led to an economic crisis. Changed behavioral patterns and a reduction of industrial activity result in a reduction in power demand, and thus Covid-19 impacts the power systems around the world. Bottom-up mapping of the effect of Covid-19 on the energy demand is challenging, if not impossible. In order to analyze the impact of the pandemic on power demand, we instead propose a simplified approach based on an econometric analysis that quantifies the country-wide load reduction of Covid-19, using the number of active cases as well as the specific lockdown period as proxies. The time span covered is from 1 January 2016 to 31 August 2020. This long time span allows us to investigate the effect of Covid-19 on the power demand. We find that in Germany (DE) and Great Britain (GB) the power demand is reduced by about 1–1.7 MW per case, while in France the demand increased by 1 MW per case during times outside of the lockdown. On the other hand, in France the lockdown itself has a much higher load reduction effect in France than in GB and DE. Based on the elasticity of power demand regarding Covid-19 cases, we calculate the impact of Covid-19 on the power prices through reduced loads. We find that Covid-19 reduced power prices by 3 to 6 EUR per MWh. The effect of Covid-19 on carbon emissions in the power sector is likely to be small. In Germany, the country with the highest absolute level, emissions in the power sector were reduced by approximately 2% (4 Mio. t $CO_2$).

**Keywords:** Covid-19; power system; carbon emissions; Europe; regression analysis





## 1. Introduction

The Covid-19 pandemic continues to be a cause of great suffering and has led to the death of more than two million people around the world by the beginning of February 2021. In order to get ahead of the pandemic during its first wave in the beginning of 2020, governments limited personal freedoms and restricted public life in a way not seen since World War II in Europe. Governments restricted freedom of movement and shut down hotels as well as restaurants. Employees were encouraged to work from home if possible. The goal was (and still is) to limit personal contacts in order to reduce the spreading of Covid-19. As a result of the anti-Covid-19 measures, economic activity decreased and an economic crisis developed. However, measuring this crisis is challenging as typical economic indicators have a great delay and only consist of a time series, so that only developments can be shown but not the causes.

The energy sector is, as most sectors, indirectly affected by the Covid-19 pandemic. On the one hand, power demand decreased due to lower production volumes in the industry and on the other hand the shift toward working from home changed the demand patterns during the first COVID-19 wave. Furthermore, the prices for fuel decreased due to lower demand on the world market (the carbon price fell shortly but returned to its previous level quickly) [1]. Power prices decreased in large European countries, like France, Great Britain, and Germany, the focus of this analysis. To what extent this reduction was caused by Covid-19 is unclear as a higher power production from renewable energies (RES)

would also reduce the power price due to the merit-order effect [2–4]. Renewable energy has a price-reducing effect in the long-term due to capacity expansion as well as in the short run in case of favorable weather conditions.

In Europe, power prices are determined by the merit-order, which orders power generation capacities according to their short-term production costs in ascending order. The last power plant needed to fulfill the demand sets the power price (for all producers and buyers) at the wholesale market [5]. As electricity can only be stored to a limited extent, generation must always exactly match demand, otherwise the energy grid collapses. Furthermore, power demand is inelastic in the short-term, so currently the balance between power generation and demand must essentially be provided by generation capacities [6]. Thus, the power demand is the critical factor to analyze the impact of Covid-19 on the energy sector.

Figure 1 presents the development of the load levels compared to the mean values of the previous years (2016–2019) for the first until the 46th calendar week for Germany, France, and Great Britain (due to the availability of the electricity sector-related data, the analyses in this paper refer to the bidding zone Great Britain (GB), constituted by England, Scotland, and Wales, and not to the United Kingdom). The calendar week number one is consistently set on the 1 January for each year. The area highlighted in green visualizes the range of minimum and maximum demand values of the previous years. It is evident that during the Covid-19 lockdown in Spring 2020 (dashed line), the demand has been considerably lower than in the previous years. In France and Great Britain, where the Covid-19 crisis in Spring 2020 was particularly severe, demand decreased up to 30%. Following the lockdown, as the pandemic slowly receded, demand increased steadily in all three countries to the initial levels. With the beginning of the fall, the second wave of the Covid-19 pandemic began and demand decreased again.

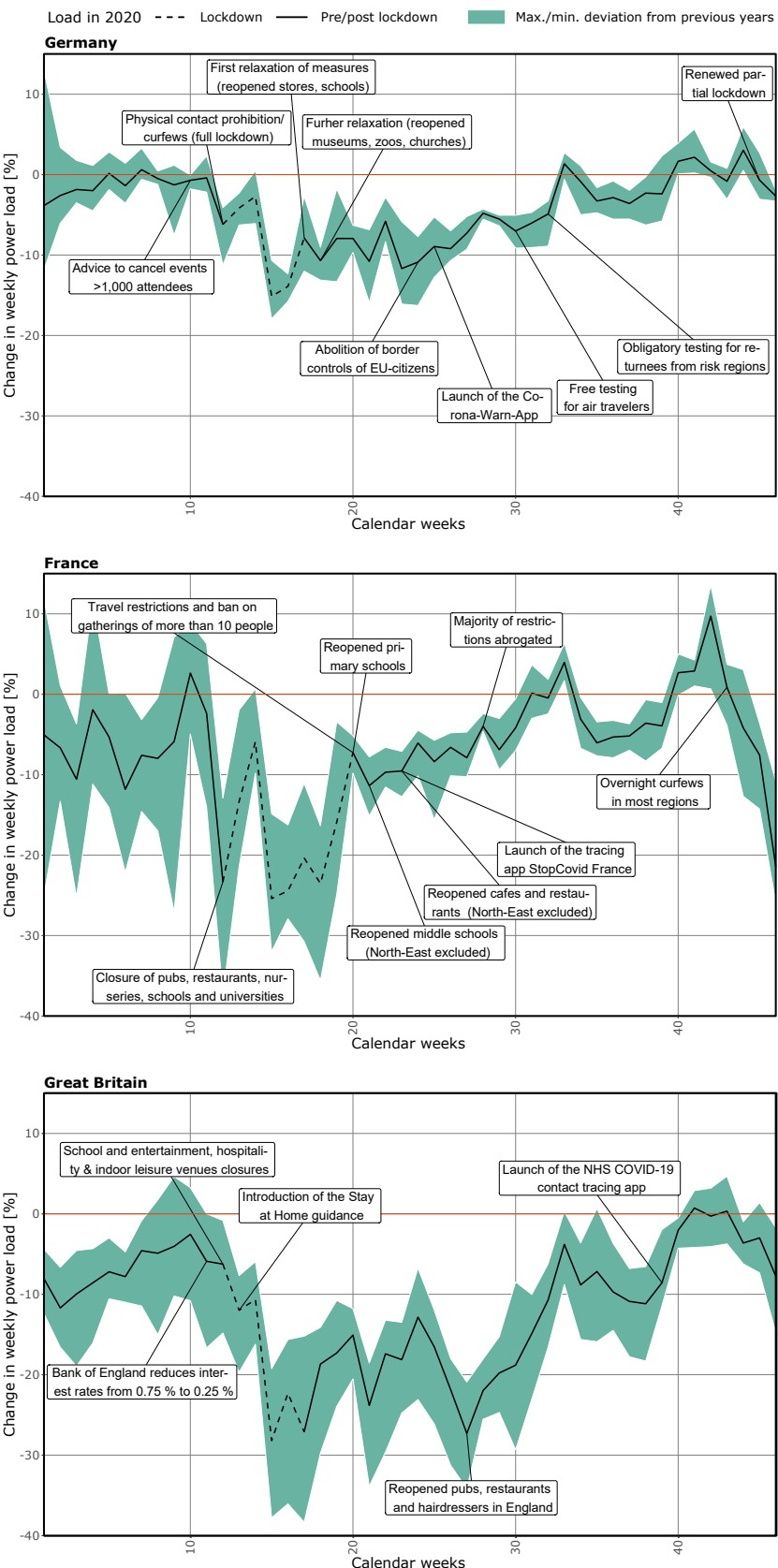

**Figure 1.** Development of the power load as a proxy for electricity demand in Germany, France, and the bidding zone Great Britain in 2020 compared to previous years (2016–2019). Load data are retrieved from in [7].

Against this background we investigate how Covid-19 affected power demand. We believe that this is an important analysis to understand the interdependence of public health and the energy economy for the case of Covid-19. This may contribute to support decision-makers and politicians in responding to pandemic challenges. Additionally, we analyze the impact of demand reductions on power prices. We base our paper on an econometric model that applies the national Covid-19 cases as an instrumental variable for the impact of Covid-19 on the electricity load. Load is thereby a proxy for demand, neglecting the utilization of power in storage within the power sector and the exchange with countries abroad. Although there is no direct connection between the Covid-19 cases and the demand for electricity, the case numbers clearly affect the power demand as policy-makers need to act when the case numbers increase, taking measures which reduce the economic activity and therefore power demand. The effect on power prices is also calculated based on the demand reduction effect of Covid-19. From the authors' perspective, it is interesting to choose the largest economies in Europe to compare their performance in the power sector during the Covid-19 pandemic. Therefore, Germany, France, and Great Britain are the countries our investigation includes. These countries are not only major economies in Europe, but also deployed different strategies to tackle the pandemic. Moreover, Germany, France, and Great Britain have disparate power generation portfolios, as outlined in [8] for 2019: Germany's power sector is mainly based on renewable energy sources (36%), lignite and coal (33%), nuclear (15%), and natural gas (11%); France's power sector is dominated by nuclear (72%) followed by renewable energy sources (19%) and natural gas (7%); and Great Britain's power sector is mainly based on natural gas (42%), renewable energy sources (26%) and nuclear power (19%), while coal only accounts only for a smaller share (2%) [8]. The diversity on the supply side of those countries allows for pointing out how different power systems deal with demand shocks.

Therefore, the paper contributes to the literature in several ways. First, it presents an approach that allows for quantifying the effect of Covid-19 on the power demand. Based on that, it increases the understanding of how energy systems deal with demand shocks and how these shocks are reflected in the power prices. Last, the effects on the transition of the energy system are analyzed.

The remainder of the paper is structured as follows. Section 2 provides an overview of the current literature and distinguishes this paper from it. Section 3 presents the applied data and methodology. Afterwards, Section 4 shows the results and discusses the energy economic context. Last, Section 5 summarizes the presented work and concludes this analysis.

## 2. Existing Literature Analyzing the Impact of the Covid-19 Crisis on the Energy Sector

There are already several publications available that analyze the impact of the pandemic on the energy sector in terms of prices and consumption (cf. [1,9–11]). The early research of Dorn et al. [12] focuses on the apparent contradiction between health insurance and economic growth in the context of the Covid-19 crisis and highlights that there are common interests in strategies on how to deal with this crisis. An extensive qualitative discussion on Covid-19 and the energy transition is given in OIES [13]. Estimates regarding short-term impacts of Covid-19 on world energy demand, $CO_2$ emissions, and investment indicators are provided in the latest World Energy Outlook [11].

In their latest World Energy Outlook from October 2020, the International Energy Agency [10] analyzes the impact of the pandemic on fuel prices, energy companies' investments, and energy demand, both on an international level as well as for selected countries and regions. Thereby, the outlook focuses on the time span between January and September 2020. Key findings are that fossil energy sources suffer more from the crisis than renewable energy sources. This can be seen especially in the developments of the global power sector, for which the outlook estimates an increase of generation from renewable energy sources and nuclear power of 272 TWh, and a drop of power from fossil fuels of 751 TWh in 2020 compared with the level in 2019. Against the backdrop of total amounts of 7167 TWh

of worldwide electricity generation from renewables, 2789 TWh from nuclear stations, and 16,951 TWh from fossil fuels, these figures are substantial. According to the outlook, there are ascertainable drops in global energy demand, e.g., the report estimates a decrease in global electricity consumption of 2–3 % in natural gas and 7 % in global coal utilization in 2020, compared to 2019 levels. For the European Union, the authors report a drop of 112 TWh in electricity demand in 2020 compared to a level of 2454 TWh in 2019. The report estimates a drop in investments in the energy sector of 18 % compared to 2019 levels.

A few further studies focus on electricity markets, in particular the German and European electricity demand as one major driver for electricity prices. During 2020, many countries have taken shut-down measures to contain the pandemic. One strand of literature focuses on the implications of these measures for energy demand (cf. Graf et al. [14], Prol and Sungmin [15]). Graf et al. [14] highlight that the Covid-19 lockdown has reduced the net demand, while controllable generation units had to react to this demand shock. In their analysis, they focus on the Italian electricity market with decreasing day-ahead prices and increasing costs for re-dispatch measurements. Prol and Sungmin [15] focus on the first wave in Europe and the USA during the spring 2020 and argue that most countries have recovered baseline demand levels by the end of July. Cicala [16] argues in the same way and finds that working from home (home office) in the United States has led to an increase in residential electricity consumption. However, both studies use a descriptive rather than a statistical approach for evaluating the interrelation between Covid-19 and electricity demand and, thus, Covid-19's effect on electricity prices. For the mid-term, Klobasa et al. [17] expect a small decrease in the German electricity demand until 2025. Afterwards, this trend may reverse due to increasing electricity demand for heating in private households and in the mobility sector.

With regard to this existing literature, the contribution of this study is twofold. First, we introduce a method to quantify the effect of Covid-19 on electricity demand. Second, we compare the actual and the counter-factual price pattern, by estimating the price reductions due to Covid-19-related demand reductions, and discuss effects for three countries: Germany, France, and Great Britain.

## 3. Methodology and Data

We derive all data related to the electricity market, i.e., load, generation of technologies, and day-ahead power market prices, from the ENTSO-E Transparency Platform [7]. The time span covered is 1 January 2016 to 31 August 2020. The source of the daily number of Covid-19-cases is Our World in Data [18]. Table 1 gives an overview of all data sources the analyses include.

**Table 1.** Time series data input to the model.

| Area | Variable | Source | Unit | Time Resolution |
|------|----------|--------|------|-----------------|
| Germany | Load, electricity generation per type | [7] | MW | Quarter-hours |
| | Day-ahead prices | [7] | €/MWh | Hours |
| | Covid-19 cases | [18] | No. of cases | Days |
| France | Load, electricity generation per type | [7] | MW | Hours |
| | Day-ahead prices | [7] | €/MWh | Hours |
| | Covid-19 cases | [18] | No. of cases | Days |
| Great-Britain | Load, electricity generation per type | [7] | MW | Half-Hours |
| | Day-ahead prices | [7] | £/MWh | Hours |
| | Covid-19 cases [1] | [18] | No. of cases | Days |
| | Exchange rate | [19] | €/£ | Days |

[1] Here, the values for the United Kingdom are used as separate values for Great Britain are not available from the used source.

In a first step of data processing for the following regression-based analyses (Section 4), the time series are harmonized. Most time series are provided in an hourly resolution. However, German load and the generation time series of production types are provided

on a quarter-hour basis. For Great Britain, the same variables are provided half-hourly. To obtain a common temporal resolution, the average values of each hour are determined for all time series with a higher than hourly resolution. For Great Britain, the day-ahead market prices in pounds are converted to euros. The source of the exchange rate is the European Central Bank [19].

Table 2 summarizes the used continuous variables. For active cases on day $d$ for country $c$, the average of new cases over the last fourteen days is used:

$$ActiveCases_d^c = \frac{1}{14} \sum_{i=0}^{13} NewCases_{d-i}^c \tag{1}$$

**Table 2.** Summary statistics.

| Statistic | N | Mean | St. Dev. | Min | Pctl(25) | Pctl(75) | Max |
|---|---|---|---|---|---|---|---|
| Load DE | 40,920 | 55,693.8 | 10,034.8 | 31,306.5 | 47,342.1 | 64,479.4 | 77,549.0 |
| Load FR | 40,854 | 53,315.7 | 11,758.3 | 29,398.0 | 44,257.5 | 61,501.0 | 94,492.0 |
| Load GB | 40,914 | 34,517.2 | 7790.6 | 511.0 | 28,574.0 | 40,195.5 | 56,478.5 |
| Price FR | 40,920 | 40.5 | 20.3 | −75.8 | 28.3 | 50.0 | 874.0 |
| Price GB (in EUR) | 40,920 | 50.7 | 23.0 | −38.8 | 39.0 | 59.6 | 1172.6 |
| Price DE | 40,920 | 34.8 | 17.1 | −130.1 | 25.9 | 44.0 | 163.5 |
| DE Biomass | 40,920 | 4547.5 | 550.3 | 0.0 | 4418.5 | 4785.9 | 5040.0 |
| DE Lignite | 40,920 | 12,995.2 | 3860.6 | 0.0 | 10,981.7 | 15,754.3 | 19,168.4 |
| DE Gas | 40,920 | 4423.9 | 2565.8 | 0.0 | 2370.8 | 6166.3 | 14,475.1 |
| DE Hard coal | 40,920 | 6798.6 | 4536.5 | 0.0 | 2720.8 | 10,224.2 | 19,143.2 |
| DE Oil | 40,920 | 320.4 | 189.3 | 0.0 | 195.7 | 439.4 | 3370.1 |
| DE Pumped storage | 40,920 | 1075.2 | 1230.0 | 0.0 | 198.9 | 1521.8 | 8217.4 |
| DE Run of river | 40,920 | 1715.9 | 419.1 | 0.0 | 1449.5 | 1956.5 | 2867.4 |
| DE Reservoir | 40,920 | 107.3 | 78.4 | 0.0 | 52.2 | 138.7 | 768.7 |
| DE Nuclear | 40,920 | 8079.8 | 1709.3 | 0.0 | 6698.3 | 9251.1 | 10,799.0 |
| DE Solar | 40,920 | 4634.0 | 7074.1 | 0.0 | 0.0 | 7455.3 | 32,947 |
| DE Waste | 40,920 | 364.7 | 120.5 | 0.0 | 319.9 | 472.6 | 568.0 |
| DE Wind offshore | 40,920 | 2155.0 | 1604.1 | 0.0 | 711.5 | 3349.8 | 6900.5 |
| DE Wind onshore | 40,920 | 9847.8 | 8224.8 | 0.0 | 3636.1 | 13,617.5 | 40,751.5 |
| FR Biomass | 40,920 | 339.6 | 91.8 | 0.0 | 292.0 | 344.0 | 665.0 |
| FR Gas | 40,920 | 3966.7 | 2471.8 | 0.0 | 2059 | 5859 | 9624 |
| FR Hard coal | 40,920 | 598.2 | 727.8 | 0.0 | 11.0 | 1058 | 2945 |
| FR Oil | 40,920 | 225.6 | 233.9 | 0.0 | 139.0 | 218.0 | 4278 |
| FR Pumped storage | 40,920 | 583.2 | 735.7 | 0.0 | 0.0 | 1038 | 3774 |
| FR Run of river | 40,920 | 4598.8 | 1605.8 | 0.0 | 3313 | 5934 | 11,430 |
| FR Reservoir | 40,920 | 1726.0 | 1057.7 | 0.0 | 937.0 | 2338 | 6143 |
| FR Nuclear | 40,920 | 42,163.6 | 8224.2 | 0.0 | 37,901 | 47,368.8 | 59,237 |
| FR Solar | 40,920 | 1157.5 | 1620.5 | 0.0 | 0.0 | 2079 | 7417 |
| FR Waste | 40,920 | 209.2 | 55.5 | 0.0 | 185.0 | 247.0 | 319.0 |
| FR Wind onshore | 40,920 | 3054.4 | 2377.7 | 0.0 | 1349 | 4026.2 | 12,976 |
| GB Biomass | 40,920 | 1145.0 | 1060.9 | 0.0 | 0.0 | 2083 | 3188 |
| GB Gas | 40,920 | 12,875.7 | 5397.9 | 0.0 | 8801.9 | 16,975.9 | 27,043.5 |
| GB Hard coal | 40,920 | 1752.3 | 2347.7 | 0.0 | 0.0 | 2414.0 | 14,658.0 |
| GB Oil | 40,920 | 0.00 | 0.3 | 0.0 | 0.0 | 0.0 | 52.0 |
| GB Pumped storage | 40,920 | 271.3 | 324.1 | 0.0 | 0.0 | 416.0 | 2180 |
| GB Run of river | 40,920 | 405.8 | 254.4 | 0.0 | 198.5 | 575.5 | 1402.5 |
| GB Nuclear | 40,920 | 6673.4 | 1329.8 | 0.0 | 6050.9 | 7587.1 | 8859.0 |
| GB Solar | 40,920 | 1240.7 | 1893.2 | 0.0 | 0.0 | 2015 | 9544 |
| GB Wind offshore | 40,920 | 1920.3 | 1433.0 | 0.0 | 751.6 | 2833.4 | 6423.1 |
| GB Wind onshore | 40,920 | 2702.7 | 1661.0 | 0.0 | 1314.0 | 3871.9 | 8340.4 |
| ActiveCases$^{DE}$ | 40,920 | 137.6 | 597.9 | 0.0 | 0.0 | 0.0 | 5121 |
| ActiveCases$^{FR}$ | 40,920 | 143.9 | 565.1 | 0.0 | 0.0 | 0.0 | 4243 |
| ActiveCases$^{GB}$ | 40,920 | 191.4 | 743.5 | 0.0 | 0.0 | 0.0 | 4725 |

## 4. Data Analysis, Results, and Discussion

### 4.1. Descriptive Analysis of Price and Load Developments

Figure 2 displays the power prices in Germany, France, and Great Britain (denoted in EUR) from 2016 to October 2020. The thick vertical lines separate the years. The thin vertical lines appear in an eight-week interval, starting from the first Monday in 2016.

All countries exhibit seasonal patterns. Concretely, summers are usually associated with lower price levels and winter months with higher levels as well as more frequent price spikes. These typical price patterns come from the higher demand due to heating (especially in France) and lower feed-in from RES (for instance, PV and hydro power plants) in winter. Within the seasonal price variations, a decline in power prices can be observed in all countries in the spring of 2020. This indicates that the Covid-19 pandemic reduces power prices as well as demand, as Figure 1 suggests.

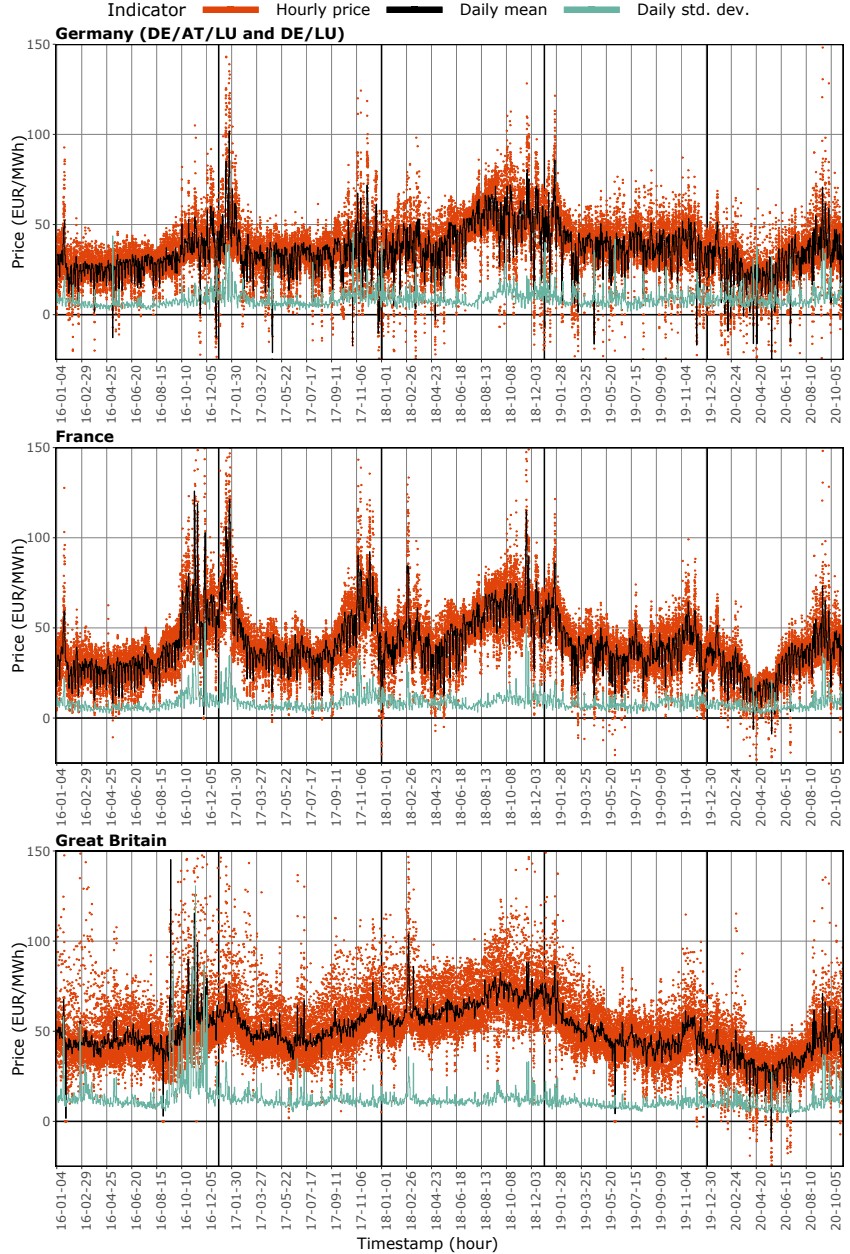

**Figure 2.** Prices in DE, FR, and GB from 2016 to 2020 (October).

*4.2. Econometric Analysis*

This section describes the two steps of this analysis. First, the load is regressed on time categorical variables and Covid-19 indicators to parse the effect of the pandemic on load in the three countries, Germany, France and Great Britain. Second, the prices of these countries are regressed on the load to identify the effect of the load changes due to Covid-19 on prices. The description and evaluation of the regressions is preceded by a preliminary analysis regarding the econometric assumptions.

4.2.1. Econometric Preliminaries

In all conducted country-specific regressions, both with load as well as prices as independent variables, positive autocorrelation and heteroskedasticity are present. This is determined by means of Durbin–Watson test and the Breusch–Pagan test (with and without normality assumption). The respective null hypotheses of homoskedastic and uncorrelated error terms are rejected at the 0.1% significance level for all regressions. Therefore, the regression results are computed and reported using robust Newey–West standard errors. All dependent variables, load, and price time series, are stationary, computed with the Augmented Dickey–Fuller test.

4.2.2. Effect of Covid-19 on Load: First Regression Analysis

First, the load of each country, $c = \{DE, FR, GB\}$, is regressed on time categorical variables in form of dummy variables. These include months (**M**), hours (**H**), years (**Y**), and days of the week (**D**). The reference categories are January, hour 12 (11:00–12:00), 2016, and Mondays. Additionally, the active Covid-19 cases are used as explanatory variables. Further, a structural variable is included in form of a dummy variable, indicating the first lockdown in the spring/summer of 2020. Last, the Covid-19 cases and the lockdown dummy variable are multiplied, providing the opportunity for the regression to compute a different effect of Covid-19 cases on the load during the lockdown. The regression follows Equation (2). Variance inflation factors (VIFs) are low in the setting of this regression as all time categorical variables are independent from each other. Naturally, the variables for Covid-19 cases and the lockdown measures correlate to a greater degree with certain months and exhibit higher VIFs.

To isolate the effect of active cases and the lockdown measure, the following regression is used. Note that in the notation of all regressions, the subscript $t$ of a matrix, e.g., $\mathbf{M}_t$, indicates that one row (for one hour) is taken from the matrix, e.g., $\mathbf{M}$:

$$Load_t^c = \beta_0 + \beta_1 ActiveCases_t^c + \beta_2 Lockdown_t^c + \beta_3 ActiveCases_t^c \cdot Lockdown_t^c + \mathbf{M}_t\boldsymbol{\mu} + \mathbf{H}_t\boldsymbol{\eta} + \mathbf{Y}_t\boldsymbol{v} + \mathbf{D}_t\boldsymbol{\delta} + \epsilon_t \tag{2}$$

Table A1 in Appendix A shows the regression results of the load analysis. The $R^2$-values are between 82% and 87%, attesting to the fact that load patterns can be explained rather well by seasonal, weekly, and diurnal patterns, in addition to long-term trends over the years. Notably, the lockdown has a substantial and statistically significant effect in Great Britain and France. France has, by far, the largest effect with a reduction in load of over 5.6 GW. These effects are smaller in Germany and Great Britain with approximately 1.3 and 2.5 GW reductions (the mean load is: 55.7 GW in Germany, 53.3 GW in France and 34.5 in GB, see Table 2). Covid-19 cases have different effects on the load of each country. In Germany and Great Britain, the effect is overall negative but to a smaller degree during the lockdown. In France, the effect is overall *positive* but to a smaller degree (almost zero) during the lockdown.

The regression results are summarized and visualized in Figure 3. In each panel, the x-axis shows the calendar weeks of 2020. The left y-axis displays the (14-day average of) active Covid-19 cases (red) as well as the *additional* load due to Covid-19 (black), which in most cases is negative. For each week, boxplots indicate the distribution of the de-seasonalized load time series. These are obtained by subtracting the effects of all time

categorical variables from the regular (observed) load time series. Therefore, every de-seasonalized load observation is expressed in the reference categories (see above).

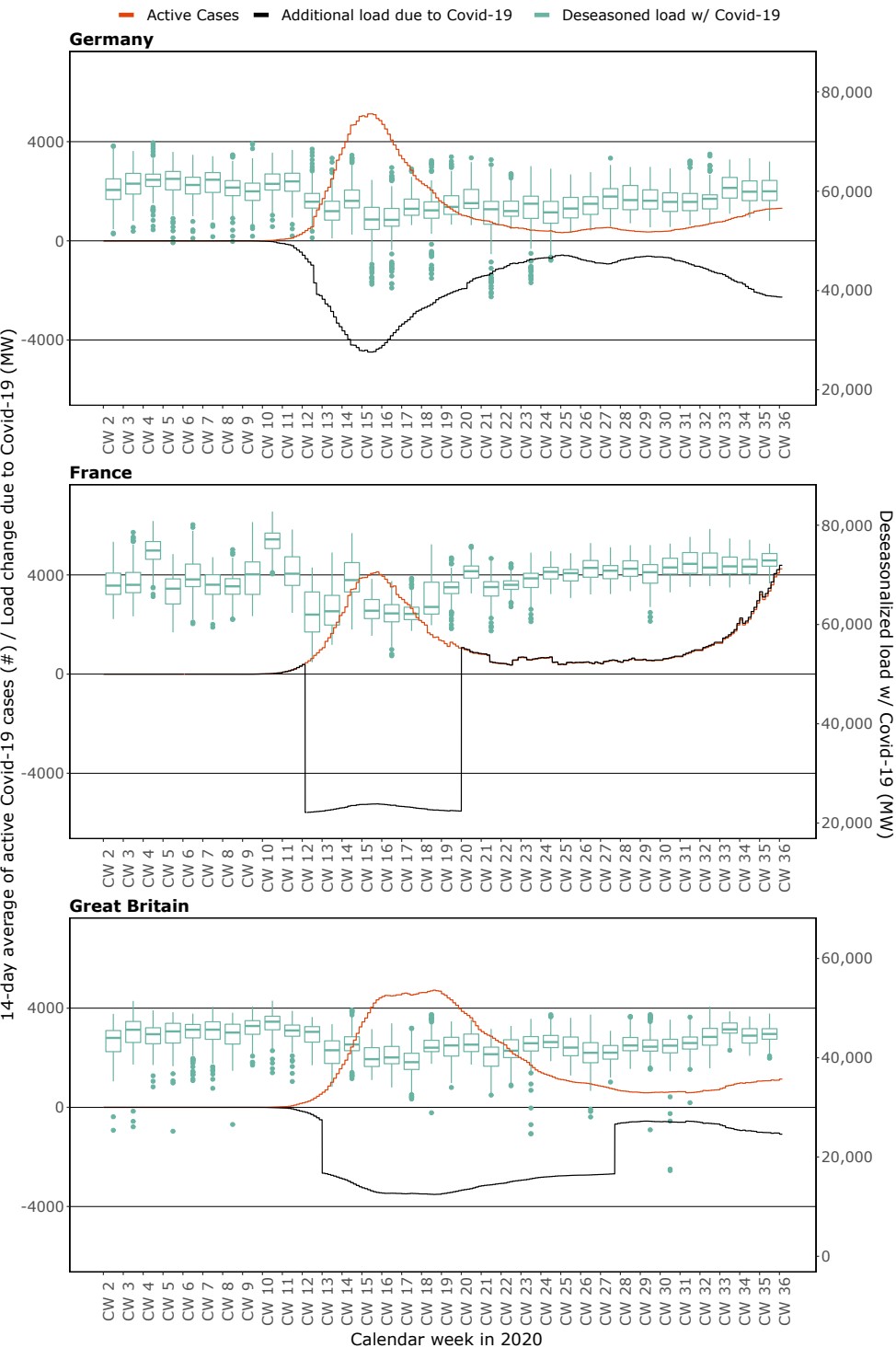

**Figure 3.** Load in Germany, France and Great Britain: Weekly distributions of de-seasonalized load with Covid-19, reduction in load due to Covid-19, and active cases.

Several insights can be derived from the regression results in Figure 3. The demand reduction due to Covid-19 can be broken down into two aspects, first the lockdown, and second the number of active cases. The lockdown variable summarizes all effects that occur during its implementation period and affect the power demand. It is therefore not surprising that in France the demand reduction is the greatest among the three countries

as the lockdown was the most rigorous. In contrast, in Germany, with its comparatively lighter lockdown, the demand reduction is the lowest. In GB, where the lockdown was imposed rather late, the effect is slightly larger than in Germany, but effective for much longer. Therefore, the absolute amount of demand reduced during the lockdown is higher in Great Britain than in Germany. The active Covid-19 cases are the second driver besides the lockdown. For this influencing factor, two different trends are observed in the three countries. In Germany and Great Britain, the active cases reduce the demand by 1.7 MW and 0.9 MW per active case, respectively. During the lockdown, this relative effect is smaller, as it only amounts 0.6 MW per active case in Germany and 0.2 MW per active case in Great Britain. Importantly, the adjusted effect of Covid-19 cases on demand during the lockdown (variable *Cases × Lockdown*) is insignificant in Germany. In contrast, in France active cases increase the demand (by about 1 MW when referring to the whole time period under investigation and roughly 0.1 MW during the lockdown). A possible explanation could be that increased working from home raises the demand in households due to the use of different electric appliances, to which residential heating likely contributes a large amount, as a great share is covered by electric heating in France. Electric heating has a much higher share in France than in UK and Germany. According to the authors of [20], households in France, despite its smaller population, consumed 56 TWh, and thus 18 TWh more electricity for space heating and warm water purposes than Germany in 2017.

The overall reduction in demand leads to a reduction in carbon emissions. The following description focuses on Germany's emissions, as its power generation portfolio features many coal-fired power plants. Based on marginal emissions of the load that Anke and Schönheit [21] calculate for 2019, Covid-19 reduced Germany's emissions by 4.4 Mt $CO_2$ which equals 2% of its emissions in the power sector. The total load reduction in Germany from January to August 2020 is 7.11 TWh due to Covid-19, estimated with Equation (2). This amount is multiplied with an average emission factor of the German power generation mix. The effect on carbon emissions is much lower in Great Britain as its power plant portfolio is based mostly on gas-fired power plants, whose emission factor is not even half the factor of coal-fired power plants (modern combined-cycle gas turbines have specific emissions in the range of 0.32 to 0.36 t $CO_2$ and hard coal power plants 0.75 to 0.95 t $CO_2$ per MWh of power output; these indications are based on emission factors of 0.201 (0.335) t per MWh of fuel input for natural gas (hard coal) according to the work in [22] and typical efficiency assumptions of 56% to 62% (35% to 45%) for combined-cycle gas (hard coal) power plants). In France, the effect on carbon emission is likely to be close to zero as power is mostly generated by $CO_2$ (almost) emission-free technologies, especially nuclear power plants (and hydro). As the load reduction due to Covid-19 is likely temporary, this emission reduction effect in Germany (and in GB) is also of temporary nature and should not be overrated. Based on current projections, further structural changes in the German power system are necessary in order to reach more ambitious long-term European climate targets, e.g., carbon neutrality in 2050. In contrast, the French power system with its large share of (near) carbon-neutral technologies appears to be already better adapted to future (climate-driven) power market conditions.

Additional to the power demand analysis, these first regression results are used as input for the following price analysis. These results are the basis for a counterfactual (abbreviated as "cf") load times series without the effect of Covid-19 cases and the lockdown, obtained by subtracting the effect of Covid-19 cases and the lockdown from the observed load time series. The computed counterfactual load time series will be used later on to compute counterfactual prices, which theoretically would have occurred without the presence of the Covid-19 pandemic.

### 4.2.3. Effect of Reduced Load on Power Prices: Second Regression Analysis

In this step, the price of each country is regressed on the dummy variables described in Section 4.2.2 as well as conventional and renewable power generation and load. As described in Section 1, renewable energy has a price-reducing effect, which necessitates

controlling for its effect on prices. Note that not all power-generating technologies are available in all countries (see Table 2), so not all explanatory variables are used for all regressions. The regressions follow Equation (3). Importantly, all load and generation variables are used in GW units instead of MW units to make the differences between countries more distinct.

To measure the effect of load on prices, the following regression is used:

$$
\begin{aligned}
Price_t^c = \beta_0 + {} & \beta_1 Load_t^c + \\
& \beta_2 Biomass_t^c + \beta_3 Lignite_t^c + \beta_4 Gas_t^c + \beta_5 Oil_t^c + \\
& \beta_6 PSP_t^c + \beta_7 RoR_t^c + \beta_8 Reservoir_t^c + \beta_9 Nuclear^c + \\
& \beta_{10} Solar_t^c + \beta_{11} Waste_t^c + \beta_{12} WindOffshore_t^c + \beta_{13} WindOnshore_t^c + \\
& \mathbf{M}_t \boldsymbol{\mu} + \mathbf{H}_t \boldsymbol{\eta} + \mathbf{Y}_t \boldsymbol{v} + \mathbf{D}_t \boldsymbol{\delta} + \epsilon_t
\end{aligned}
\tag{3}
$$

Multi-collinearity is generally low in these regressions, as most continuous variables exhibit VIFs of below 5. The exceptions include load and solar generation, which is unsurprising because load is regressed on the included dummy variables in Section 4.2.2 and shows high $R^2$-values. Similarly, solar generation follows seasonal and diurnal patterns, so much of its variation is explained by the dummy variables.

Table A2 in Appendix A displays the regression results. The $R^2$-values are lower than the ones of the load regressions. For Germany and France, well over 70% of the price variance can be explained by the included independent variables. For Great Britain, the $R^2$-value is closer to 40%. Great Britain is dominated to a larger extent by gas-fired power plants. This likely makes gas the price-setting technology during more hours compared to France or Germany, which introduces a greater price variance into the power system. This is confirmed by the greater standard deviation in Table 2 for Great Britain (~23 EUR/MWh compared to approximately 20 and 17 EUR/MWh for France and Germany, respectively). Therefore, explaining price variance in Great Britain is a more difficult task for the regression.

The effect of load on prices is very similar in all countries. The country-specific effects range between 0.952 EUR/GW and 1.075 EUR/GW. As expected, load has a positive effect on prices. From Figures 1 and Figure 3, it can be observed that Covid-19 generally reduced the load in the analyzed countries. Therefore, power prices would have been higher, on average, in the absence of the Covid-19 pandemic. This means that wholesale power prices are reduced by about 3 EUR/MWh in GB, 4 EUR/MWh in DE, and about 6 EUR/MWh in France during the lockdown. The lower wholesale power prices likely have a limited effect on the energy industry because they do not generate any operational losses, as power plants only operate when they have a positive margin. Therefore, only the coverage of fixed costs is affected. Thus, in light of the long power plant life times, the impact of this short period is very limited. To sum up, the impact of Covid-19 is likely to be very limited on the power plant operators.

The non-observed, counterfactual prices can be computed by means of Equation (3) and the counterfactual load time series created in Section 4.2.2. The factual are replaced by the counterfactual load time series and prices are computed by means of the regression coefficients, shown in Table A2.

Figure 4 depicts three different German power prices for each hour of calendar week 15 in 2020. The gray, solid dots indicate the real power price, i.e., the power prices that are observed and used to compute the regression results according to Equation (3). The red boxes indicate the regression's estimates of the dependent variable, namely, the price. The difference between the gray solid dots and the red boxed is the error made by the regression in estimating the prices, namely, the residuals of the regression. The green X's are the estimated price when using the counterfactual load time series ("estimated cf"). The differences between the red boxes and green X's are marked by solid lines. This can be interpreted as the price effect due to Covid-19. As stated above, these counterfactual prices are higher in the absence of Covid-19.

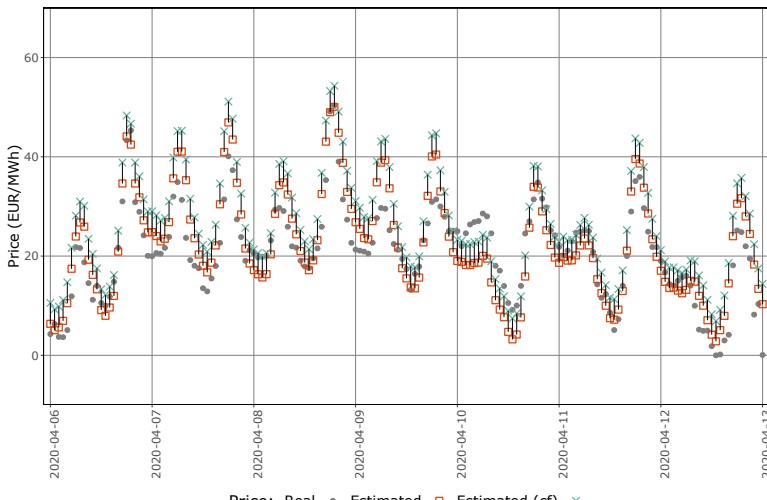

**Figure 4.** Prices in Germany in CW 15 2020, for the factual and counterfactual case, i.e., with and without Covid-19.

For the two power prices (estimated factual and estimated counterfactual), Figure 5 depicts the price distributions for all three countries. An interesting insight from this table is the shift of the distributions to the right when factoring out the Covid-19 effect. Concretely, higher price categories are populated more often and lower price categories less often in the counterfactual case.

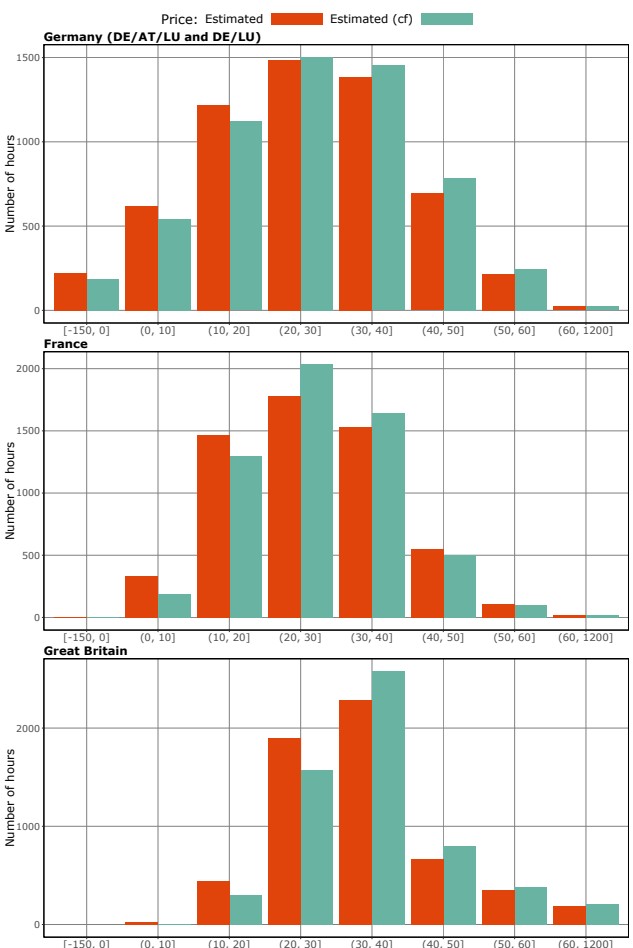

**Figure 5.** Price distributions in Germany, France, and Great Britain in 2020, for the factual and counterfactual case, i.e., with and without Covid-19.

### 4.3. Limitations of the Proposed Approach

Using the number of active Covid-19 cases as a proxy for the reduction of economic activity has some drawbacks. First, economic activity is reduced due to political decisions that are based on the current Covid-19 cases but also on the potential threat of Covid-19. This relationship is only partly reflected in the number of active cases. Furthermore, a linear relationship between number of cases and the economic activity, and therefore the power demand, simplifies the real context. However, applying the assumptions allows to quantify the impact of Covid-19 and to put it into perspective, despite a high uncertainty in the results.

Another limitation concerns the quality of the data provided by the ENTSO-E. The ENTSO-E Transparency platform is widely used in academic literature and provides the data very promptly, which is why it has been used. However, it exhibits certain data quality issues. For an investigation with regard to the data quality of the works in [7], the authors refer to the analysis conducted by Hirth et al. [23].

## 5. Conclusions and Policy Implications

In response to Covid-19, many governments have severely restricted public life, which has led to a decline in economic activity, among other things. In the energy sector, this results in a decrease in demand. Other effects of Covid-19 on the energy industry, such as changes in electricity prices or $CO_2$ emissions, can be attributed at least partly to the demand reduction. In order to analyze the effects of Covid-19 in detail, it is therefore essential to understand the decline in demand.

This paper develops a methodology that attributes the demand reduction to two aspects of the Covid-19 crisis: first, the lockdown periods, and second the active case numbers. Both aspects serve as instrumental variable in a regression analysis. We show that a rigorous lockdown, as, for instance, in France, decreased the load significantly. Prompt political reactions to the Covid-19 crisis went along with a rapid decline in demand, but also a fast subsequent increase to pre-lockdown levels. By contrast, Great Britain's late reaction to Covid-19 led to a slower recovery of electricity demand. In Germany and Great Britain, the active cases reduce the demand by 1.7 MW and 0.9 MW per active case, which seems to be a very high figure per case and can only be explained by severe restrictions, especially for industry and commerce resulting in a decline in electricity demand. In contrast, in France active cases increase the demand (by about 1 MW) when referring to the whole time period under investigation and roughly 0.1 MW during the lockdown. Where this difference between France and the two other countries comes from can only be guessed at this point. One explanation could be the higher share of electric heating in France, which has overcompensated for the decline in industrial demand due to the higher heating demand associated with home office activities (while this demand is satisfied by other heating technologies in the two other countries). Furthermore, the progress of the Covid-19 crisis can only be compared to a limited extent on the basis of different policy measures and their impact on demand. Further aspects, e.g., different social norms, housing situations, etc., affect the development of the crisis. The decreases in demand affected the utilization of the power plants in all analyzed power systems. Although the power generation levels and fleet structures differ greatly throughout the examined areas, in all the countries the demand shocks were handled well, as there were no blackouts or substantial deviations from the normal grid frequency during 2020 (to the knowledge of the authors). Furthermore, the lower demand reduced power prices by about 1 EUR/GW in all three countries. This short-term drop in prices should not have an excessive long-term impact on the financial situation of power plant operators. It only affects the recovery of fixed costs as the operational expenses are reduced due to lower running times.

However, it is likely that existing trends, such as the coal phase-out, will be accelerated due to Covid-19, resulting in lower electricity and gas price levels and relatively stable emission allowance prices. As the framework conditions for operating coal-fired power plants become (even) less attractive, this may result in an earlier phase-out. The reason is that $CO_2$ prices increased beyond pre-Covid-19 levels, even if emission prices decreased in the short-term during the crisis. The strong political volition to decarbonize is changing market expectations and would make carbon prices rise, although fundamental market data would have suggested falling prices due to lower demand for emission allowances. This may be a signal that long-term market expectations foresee a shortage of allowances in emission trading.

Our results show surprising insights, as it could be expected that Covid-19 would lead to a larger effect of demand reduction on $CO_2$ emissions. Beyond the changes in $CO_2$ prices, Covid-19 also provides one-time savings in $CO_2$ emissions in the energy industry and also in the end-use sectors due to the restrictions for companies and people. These savings in the power sector are particularly high in Germany (~4.4 Mt $CO_2$), as a large share of electricity is generated by coal-fired power plants. In Great Britain, the effect is significantly lower, as natural gas-fired power plants account for the majority of conventional generation. In France, however, the expectation would be that there will be very low $CO_2$ savings, as $CO_2$ emission-free technologies, such as nuclear power plants, account for the majority of electricity generation.

Future works could compare the second lockdown in the winter 2020/21 with the lockdown during the spring of 2020. It can be expected that results would differ, not only for seasonal differences in power demand, but also because of altered strategies how politics deal with the pandemic during the second wave. Moreover, it would be also interesting to compare other non-EU countries, e.g., Russia, China, or the USA, and their performance during the crisis. However, due to non-availability of data, this was not an option for this research. It will be possible for researchers with access to data of our analyzed countries to reproduce our methodology.

**Author Contributions:** Conceptualization, all; methodology, D.S.; Validation, all; formal analysis, D.S. and H.S.; investigation, P.H., H.S., C.-P.A., and D.M.; data curation, D.S. and H.S.; writing—original draft preparation, all; Writing—review and editing, all; visualization, D.S. and H.S.; supervision, D.M.; project administration, P.H., H.S., and C.-P.A. All authors have read and agreed to the published version of the manuscript.

**Funding:** Open Access Funding by the Publication Fund of the TU Dresden.

**Institutional Review Board Statement:** Not Applicable.

**Informed Consent Statement:** Not Applicable.

**Data Availability Statement:** Not Applicable.

**Conflicts of Interest:** The authors declare no conflicts of interest.

## Appendix A

**Table A1.** Regression results: Load regressed on dummy variables and active Covid-19 cases.

| | Dependent Variable: | | | | | |
|---|---|---|---|---|---|---|
| | **Load Germany** | | **Load France** | | **Load Great Britain** | |
| M2 | 422.194 | (340.876) | −2888.381 *** | (684.809) | 456.160 ** | (202.678) |
| M3 | −1922.561 *** | (381.430) | −9201.241 *** | (674.498) | −1012.213 *** | (226.877) |
| M4 | −5756.182 *** | (413.453) | −18,005.280 *** | (628.011) | −4582.063 *** | (226.007) |
| M5 | −7163.971 *** | (433.436) | −23,416.520 *** | (572.624) | −6562.748 *** | (210.444) |
| M6 | −6650.721 *** | (384.556) | −24,319.020 *** | (524.629) | −7558.618 *** | (200.347) |
| M7 | −6344.384 *** | (332.092) | −23,540.720 *** | (526.957) | −8165.849 *** | (204.330) |
| M8 | −6932.539 *** | (346.567) | −26,765.220 *** | (570.881) | −8207.455 *** | (198.625) |
| M9 | −6147.534 *** | (344.531) | −23,936.380 *** | (532.576) | −7686.319 *** | (199.070) |
| M10 | −4537.314 *** | (407.242) | −20,147.620 *** | (605.574) | −6135.953 *** | (204.812) |
| M11 | −957.744 ** | (392.039) | −9065.915 *** | (703.873) | −2338.358 *** | (233.750) |
| M12 | −2729.572 *** | (650.070) | −4258.198 *** | (724.622) | −2240.949 *** | (296.223) |
| H0 | −17,501.700 *** | (136.086) | −8279.654 *** | (105.358) | −12,620.410 *** | (78.424) |
| H1 | −17,834.260 *** | (132.249) | −10,217.430 *** | (105.724) | −13,465.900 *** | (73.917) |
| H2 | −17,260.640 *** | (124.682) | −11,730.220 *** | (96.245) | −14,442.040 *** | (70.230) |
| H3 | −15,467.900 *** | (107.851) | −11,360.250 *** | (88.869) | −15,037.21 0 *** | (68.327) |
| H4 | −11,332.950 *** | (101.137) | −8971.546 *** | (85.915) | −14,833.850 *** | (72.368) |
| H5 | −6455.908 *** | (96.711) | −5351.098 *** | (90.315) | −11,873.430 *** | (92.577) |
| H6 | −2621.457 *** | (90.459) | −2107.347 *** | (94.882) | −7111.801 *** | (99.300) |
| H7 | −604.753 *** | (73.903) | −139.798 * | (84.505) | −3297.327 *** | (86.484) |
| H8 | 816.413 *** | (61.583) | 843.981 *** | (67.017) | −1410.060 *** | (66.999) |
| H9 | 2085.811 *** | (55.466) | 1356.210 *** | (51.341) | −481.701 *** | (50.862) |
| H10 | 2197.305 *** | (42.290) | 2001.198 *** | (44.487) | 96.916 *** | (35.223) |
| H11 | 1276.913 *** | (32.008) | 1406.062 *** | (34.727) | 379.551 *** | (32.669) |
| H13 | −1121.775 *** | (31.062) | −1736.809 *** | (34.049) | −619.468 *** | (32.520) |
| H14 | −1978.539 *** | (39.318) | −3279.472 *** | (41.560) | −974.399 *** | (39.925) |
| H15 | −1857.239 *** | (51.425) | −3893.380 *** | (47.165) | −293.812 *** | (48.445) |
| H16 | −798.615 *** | (72.155) | −2470.932 *** | (55.956) | 1257.548 *** | (61.391) |
| H17 | −497.028 *** | (83.496) | −156.780 ** | (71.163) | 1966.567 *** | (74.715) |
| H18 | −1933.916 *** | (92.341) | −247.630 *** | (92.702) | 1464.667 *** | (79.155) |
| H19 | −4693.753 *** | (90.959) | −2121.782 *** | (79.541) | 191.572 *** | (72.975) |
| H20 | −7442.875 *** | (102.887) | −2612.972 *** | (74.414) | −1862.246 *** | (65.218) |
| H21 | −10,535.660 *** | (121.919) | −1907.874 *** | (82.858) | −4936.481 *** | (63.074) |
| H22 | −13,822.410 *** | (125.051) | −3585.080 *** | (93.476) | −8537.473 *** | (62.786) |
| H23 | −16,142.250 *** | (123.700) | −6376.530 *** | (97.984) | −11,655.790 *** | (67.817) |
| Y17 | 765.258 *** | (218.170) | 114.840 | (298.513) | −2729.682 *** | (120.104) |
| Y18 | 1507.393 *** | (222.966) | −562.954 * | (318.924) | −2383.713 *** | (121.124) |
| Y19 | 541.544 ** | (231.578) | −993.105 *** | (286.178) | −3132.017 *** | (128.941) |
| Y20 | −956.817 *** | (360.718) | −3896.747 *** | (397.956) | −4610.393 *** | (229.821) |
| Sun | 8166.436 *** | (197.964) | 5085.015 *** | (257.860) | 2742.593 *** | (119.157) |
| Tue | −3718.844 *** | (149.147) | −2253.437 *** | (242.741) | −674.786 *** | (120.558) |
| Wed | 7827.980 *** | (253.729) | 4449.022 *** | (299.856) | 2568.842 *** | (146.569) |
| Thu | 9533.097 *** | (211.792) | 5739.352 *** | (303.055) | 3156.884 *** | (131.226) |
| Fri | 9803.277 *** | (199.441) | 6012.438 *** | (300.013) | 3187.811 *** | (131.571) |
| Sat | 9399.969 *** | (213.165) | 5885.513 *** | (298.368) | 3134.739 *** | (134.830) |
| LD | −1291.138 | (944.069) | −5622.362 *** | (1236.838) | −2529.089 *** | (438.879) |
| C | −1.728 *** | (0.587) | 1.034 *** | (0.217) | −0.948 ** | (0.410) |
| C · LD | 1.105 | (0.675) | −0.940 * | (0.501) | 0.740 * | (0.432) |
| Const. | 59,648.090 *** | (358.467) | 69,702.430 *** | (576.102) | 44,583.620 *** | (200.996) |
| Adj. R$^2$ | 0.8233 | | 0.8420 | | 0.8699 | |

Note: Standard errors in parentheses; C: Cases, LD: Lockdown; * $p < 0.1$; ** $p < 0.05$; *** $p < 0.01$.

**Table A2.** Regression results: Price regressed on market variables.

| | Price Germany | | Dependent Variable: Price France | | Price Great Britain | |
|---|---|---|---|---|---|---|
| Load | 0.952 *** | (0.059) | 1.075 *** | (0.042) | 1.039 *** | (0.087) |
| Biomass | −0.128 | (0.684) | −11.486 *** | (2.120) | −0.711 * | (0.429) |
| Lignite | 0.782 *** | (0.102) | | | | |
| Gas | −0.138 | (0.138) | 0.752 *** | (0.116) | −0.095 | (0.078) |
| Hard coal | 0.109 | (0.078) | 1.201 *** | (0.354) | 1.065 *** | (0.171) |
| Oil | 7.379 *** | (1.355) | 18.509 *** | (2.485) | 128.143 | (95.971) |
| Pumped storage | 1.057 *** | (0.103) | 2.838 *** | (0.242) | 12.584 *** | (1.197) |
| Run of river | −6.125 *** | (0.611) | −3.283 *** | (0.201) | 3.771 *** | (1.191) |
| Reservoir | 1.574 | (1.798) | −0.545 * | (0.293) | | |
| Nuclear | −0.447 *** | (0.168) | −0.027 | (0.035) | −1.339 *** | (0.222) |
| Solar | −0.832 *** | (0.040) | −2.091 *** | (0.131) | −0.958 *** | (0.094) |
| Waste | 10.988 *** | (2.005) | 25.638 *** | (4.001) | | |
| Wind offshore | −0.313 ** | (0.122) | | | −1.867 *** | (0.184) |
| Wind onshore | −0.936 *** | (0.051) | −1.324 *** | (0.081) | −0.652 *** | (0.164) |
| M2 | −2.737 *** | (0.936) | 1.526 | (0.962) | −3.294 *** | (1.055) |
| M3 | −3.107 *** | (0.914) | 5.911 *** | (0.937) | −3.367 *** | (0.956) |
| M4 | −0.958 | (1.029) | 13.104 *** | (1.101) | 0.436 | (1.230) |
| M5 | 1.609 | (1.206) | 20.004 *** | (1.284) | 2.766 ** | (1.340) |
| M6 | 5.831 *** | (1.206) | 22.432 *** | (1.337) | 2.528 * | (1.389) |
| M7 | 7.511 *** | (1.081) | 20.290 *** | (1.256) | 4.909 *** | (1.357) |
| M8 | 6.891 *** | (1.064) | 21.307 *** | (1.362) | 5.798 *** | (1.310) |
| M9 | 5.769 *** | (1.018) | 18.755 *** | (1.397) | 10.191 *** | (2.078) |
| M10 | 4.037 *** | (1.096) | 21.558 *** | (1.281) | 7.827 *** | (1.390) |
| M11 | 3.185 *** | (0.942) | 16.734 *** | (1.365) | 9.363 *** | (1.721) |
| M12 | 3.278 *** | (1.064) | 9.148 *** | (0.997) | 6.213 *** | (1.128) |
| H0 | 1.280 | (0.789) | −3.987 *** | (0.566) | 8.743 *** | (1.010) |
| H1 | 0.529 | (0.792) | −4.412 *** | (0.591) | 7.900 *** | (1.067) |
| H2 | −0.288 | (0.762) | −4.781 *** | (0.615) | 6.252 *** | (1.131) |
| H3 | −0.938 | (0.694) | −4.408 *** | (0.609) | 5.822 *** | (1.170) |
| H4 | −0.828 | (0.547) | −2.362 *** | (0.577) | 7.304 *** | (1.157) |
| H5 | −0.113 | (0.412) | −0.184 | (0.510) | 9.439 *** | (0.951) |
| H6 | 1.182 *** | (0.306) | 1.364 *** | (0.432) | 7.009 *** | (0.639) |
| H7 | 1.565 *** | (0.249) | 1.609 *** | (0.335) | 5.853 *** | (0.463) |
| H8 | 1.542 *** | (0.213) | 1.477 *** | (0.242) | 6.864 *** | (0.390) |
| H9 | 1.491 *** | (0.215) | 2.034 *** | (0.177) | 5.488 *** | (0.300) |
| H10 | 1.246 *** | (0.187) | 1.315 *** | (0.141) | 3.505 *** | (0.204) |
| H11 | 0.567 *** | (0.126) | 0.557 *** | (0.105) | 1.765 *** | (0.133) |
| H13 | 0.139 | (0.123) | 0.152 | (0.117) | −1.464 *** | (0.116) |
| H14 | 0.127 | (0.195) | 0.998 *** | (0.196) | −1.646 *** | (0.171) |
| H15 | 0.564 ** | (0.255) | 2.665 *** | (0.282) | 1.061 *** | (0.252) |
| H16 | 1.059 *** | (0.316) | 4.811 *** | (0.357) | 6.622 *** | (0.402) |
| H17 | 1.248 *** | (0.363) | 6.919 *** | (0.931) | 15.466 *** | (1.032) |
| H18 | 0.745 ** | (0.377) | 3.620 *** | (0.526) | 13.262 *** | (0.620) |
| H19 | 0.283 | (0.409) | 2.339 *** | (0.482) | 6.757 *** | (0.547) |
| H20 | 0.438 | (0.474) | 0.959 * | (0.496) | 2.138 *** | (0.520) |
| H21 | 0.680 | (0.554) | −2.120 *** | (0.499) | 0.882 | (0.579) |
| H22 | 0.916 | (0.665) | −2.854 *** | (0.520) | 6.929 *** | (0.774) |
| H23 | 1.386 * | (0.746) | −3.068 *** | (0.547) | 8.480 *** | (0.946) |
| Y17 | 5.420 *** | (0.470) | 4.672 *** | (0.596) | 7.414 *** | (0.604) |
| Y18 | 14.275 *** | (0.607) | 18.036 *** | (0.601) | 22.484 *** | (1.001) |
| Y19 | 11.060 *** | (0.925) | 8.354 *** | (0.527) | 8.483 *** | (1.244) |
| Y20 | 7.367 *** | (1.264) | 5.286 *** | (0.611) | 0.052 | (1.552) |
| Sun | −1.955 *** | (0.550) | −0.242 | (0.431) | −1.472 *** | (0.469) |
| Tue | −1.079 ** | (0.488) | −2.546 *** | (0.407) | −0.860 * | (0.467) |
| Wed | −2.709 *** | (0.664) | −0.600 | (0.571) | −0.196 | (0.825) |
| Thu | −2.550 *** | (0.622) | −0.607 | (0.514) | −1.086 ** | (0.551) |
| Fri | −2.396 *** | (0.626) | −0.707 | (0.496) | −1.329 ** | (0.556) |
| Sat | −2.397 *** | (0.611) | −0.742 | (0.507) | −1.071 | (0.659) |
| Constant | −16.632 *** | (4.079) | −25.083 *** | (3.172) | 9.283 ** | (4.508) |
| Adjusted $R^2$ | 0.7760 | | 0.7298 | | 0.4583 | |

Note: Standard errors in parentheses; * $p < 0.1$; ** $p < 0.05$; *** $p < 0.01$.

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
