# Peer review of "Covid-19’s Impact on European Power Sectors: An Econometric Analysis"

_energies, doi:10.3390/en14061639_

Round 1

Reviewer 1 Report

The paper is well written, based on new data, and fits the special issue well.

References to sources in the text are missing, they need to be inserted in proper (readable) format. Additionally, the paper lacks list of references.

A section should not end with figures, some text should follow.

Author Response

please find the response attached

Reviewer 2 Report

Overall, an interesting study. I have a wide variety of suggestions large and small.

Please double-check your details. For example: 

  • In the Introduction you write that thousands of died from Covid-19 — however it actually is millions
  • You write that Germany has a high share of energy from coal. It's actually only about a quarter coal — and that's about the same percentage as wind-power generation. This also means that you need to find another explanation for the difference between UK and Germany at the bottom of page 9 and info about Germany in the conclusion and elsewhere. Frankly, instead of simply saying "large share," you should add a paragraph in the introduction that breaks down power generation by type for all of the countries you studied. 
  • You write that energy prices may have dipped because of increased production from renewable sources. Did renewable-energy production increase during the time studied? 

Please ensure that verb tenses are consistent. For example, sentences switch back and forth from past to present tense in the Introduction. 

You write that "The countries Germany, France and Great Britain are the focus of this investigation" -- and while you explain why those countries work for you, you do not explain why you have chosen those and not others. For example, why not Italy, Spain or Russia? You should state why you chose those three explicitly as opposed to others. 

I'm guessing that you've already been advised by others who have read your earlier drafts that you need to address the "so what" of your work, especially because the result, arguably, is obvious. However in my judgment you need to do a better job at sharpening your explanation of why this work matters in the introduction and particularly in the conclusion. 

Lastly, I appreciate the annotations on Fig. 1. 

Author Response

Please find the response attached.

Reviewer 3 Report

Dear Authors

I like your manuscript: the topic is current, the text is well written, and three countries with different energy mixes are well chosen. The conclusions are accurate and well-balanced.

In my opinion, the results are surprising - everyone expected a greater reduction in CO2 emissions due to the Covid lockdown.

Good job!

Author Response

Please find the response attached.

Round 2

Reviewer 2 Report

Thank you for the the improved draft. However there were a few concerns that you did not address in the paper: 

  • You write that "Germany has a high share of coal capacities, while France has many nuclear plants and the Great Britain has a lot of gas-fired power stations." You explained to me that you did this because you say that Germany is 33% powered by coal. Explaining it to me is insufficient -- it needs to be the in the paper itself, and cited. Instead of simply saying "large share," you should add a paragraph in the introduction that breaks down power generation by type for all of the countries you studied.
  • You explained to me why you chose UK, France and Germany and not others, such as Russia or Italy — but you should explain this in the paper itself. 
  • It remains the case that in my judgment you need to do a better job at sharpening your explanation of why this work matters in the introduction and particularly in the conclusion. You made no changes in this regard, but you should. 

Round 3

Reviewer 2 Report

Thank you for your successful efforts to improve the article!

From my perspective, there is one small thing to correct before publication: I suggest asking a colleague who is a native English speaker if s/he can edit the article for you. The article has small grammatical errors throughout. (For example, "We belief" should be "We believe")

Overall: Well done! Let's just get the grammar fixed. 
